# Preventing School Exclusion of Students with Autism Spectrum Disorder (ASD) through Reducing Discrimination: Sustainable Integration through Contact-Based Education Sessions

**Gheorghița Nistor [1,*] and Cristian-Laurențiu Dumitru [2]**

[1] Department of Social Work, Faculty of Sociology and Social Work, University of Bucharest, 050663 Bucharest, Romania

[2] SucCESS Association, Support for Children with Special Educational Needs and Their Families, 917035 Calarasi, Romania; asociatia.succes@gmail.com

\* Correspondence: gheorghita.nistor@unibuc.ro; Tel.: +40-0722-464-111

**Abstract:** Students with autism spectrum disorder (ASD) are discriminated against and stigmatized by the school community. The objective of this study is to analyze the school inclusion process of students with ASD by reducing discrimination and stigmatization through contact-based education sessions. This can be achieved through school projects. In the present study, discrimination and stigmatization toward children with ASD were analyzed in high school students (N = 141) through Haghighat's standardized stigmatization questionnaire (SSQ1). In the active group, a student diagnosed with ASD also participated in the awareness activities of the problems faced by the students with ASD and the contact-based education (CBE) sessions carried out in the classroom. The results showed significant differences in reducing discrimination and stigmatization in high school students, both in the control group and especially in the active group. It was observed that the development of CBE (inclusion of the student with ASD in activities) led to the creation of a supportive school community, demonstrating that the activities carried out within the SucCESS Project achieved their purpose. The SSQ1 can be applied to high school students, and together with CBE activities, it can be used in school inclusion projects for children with ASD or mental disabilities.

**Keywords:** autism spectrum disorder; disabilities; discrimination; standardized questionnaire; contact-based education; sustainable school inclusion

## 1. Introduction

### 1.1. Conceptual Framework

Autism is the most common disorder in a wider range of developmental disorders (high-functional autism, Asperger's syndrome, Rett syndrome, etc.); therefore, the notion of autism spectrum disorder (ASD) is frequently used. Using the typology developed by the World Health Organization, International Statistical Classification of Diseases and Related Health Problems [1], autism is a developmental mental and behavioral disorder that affects three areas of functioning: social interaction, communication and behavior. The most common symptoms are repetitive actions, communication difficulties and limited interactions. The disorder can be controlled with medication to control attention deficit, aggression or stereotypical behaviors, but the most important intervention is therapy, which should improve communication skills and reduce behavioral disorders [2].

Currently, worldwide, there are estimated to be approximately 52 million cases diagnosed with ASD, meaning about 1–2% of children worldwide [3]. In the US alone, in 2018–2019, over 8 million children aged 3–17 had a mental or behavioral health problem. Anxiety, behavioral disorders and depression are the most commonly diagnosed disorders [4]. It is difficult to get a clear image of global prevalence, as there are substantial variations in current country estimates. Although there are these variations in estimates,

studies have not shown that the variability of ASD can be explained by geographical, ethnic, cultural or socioeconomic factors [5]. Depression, which is one of the most common mental illnesses, is commonly associated with ASD. If one in four people goes through at least one depressive episode at some point in their life, then within people with ASD, depression and discrimination are common problems [6].

The study of this pathology has aroused the interest of specialists especially with the development of technologies that have been used successfully in effective interventions in people diagnosed with ASD. Robotic devices and mobile applications have been used to acquire cognitive, social and relational skills. An analysis of the literature (especially Web of Science) showed that after 2009, the volume of research increased considerably highlighting the advantages of using technologies in the recovery and therapeutic processes of people diagnosed with ASD [7].

### 1.2. Discrimination and Stigmatization of Schoolchildren with Autism

The objective of this study is to analyze the school inclusion process of students with ASD by reducing discrimination and stigmatization through contact-based education sessions. This can be achieved through school projects.

Children with ASD are forced to adapt to social life without being able to use the adaptive elements of typical children. They encounter difficulties on several existential levels, and therefore educational projects aimed at school integration and optimizing the environment for these students must include as many of them as possible [8]. Sometimes teachers prefer these children to attend special schools or question the benefits of the school inclusion process for students with disabilities, because they have to make a series of changes at the curricular, social or behavioral level [9,10]. There is a direct relationship between the architecture of spaces, classrooms or other school activities and the academic motivation, well-being or social relationships of students. The relationship between school spaces and the socioeducational well-being of the student is a challenge for schools that have to manage the process of educational inclusion of students with disabilities [11]. Other authors pointed out that teachers can have favorable attitudes toward the school inclusion of students with disabilities if they participate in the decision-making process regarding the inclusion of these students in their classes [12]. In the same time, social environment has a great impact on human beings, and by communicating, one can feel accepted and understood [13].

The timing of the diagnosis of ASD has a strong impact on the family. From the moment of diagnosis, the quality of family life is affected by this disorder, whether we are talking about the health of family members, their personal and social life, relationships within the family or with the extended family or financial efforts necessary for therapy. An ecological approach to individual and environmental factors on the quality of family life may be relevant to understanding what parental resources should be targeted in recovery and adaptation programs for the school and social community of the child with ASD [14–17].

A family with a child with ASD experiences high levels of stress over long periods of time, far exceeding the level of stress in a family with neurotypical children or even in families whose children have been diagnosed with pathologies such as down syndrome, cerebral palsy, mental retardation or cystic fibrosis [18–20]. Other stressors for parents and therapists, in daily management, may be the atypical eating behaviors (food selectivity) manifested by these children, which become targets that are difficult to overcome [21]. During the pandemic caused by COVID-19, adapting to new routines was a major challenge, with a psychological impact. Parental stress and the emotional well-being of children with ASD have been unfavorably affected by social isolation [22,23]. After 2005, it has been observed that these aspects, stress and family problems, the way in which the relations between the parents are affected, aroused the interest of the scientific community, a fact noticed by the large number of studies published in Web of Science. How parents

relate to the diagnosis received by the child influences the evolution of the therapy and socioeducational integration process [24].

People diagnosed with mental disorders are discriminated against, are stigmatized [25,26], and can be victims of violence, aggression, and bullying. The effects of bullying are serious on children who fall victim to this phenomenon, ranging from depression, fear, anxiety, relationship difficulties, behavioral disorders, exclusion, and suicidal thoughts. Bullying and aggressive behaviors are more common in children with ASD than in typical children [27]. The social manifestation of the feeling of shame can prevent those who are potential victims from obtaining the support they need to recover and actively participate in the life of the community. Stigma is one of the major sufferings of people with mental disabilities, affecting their quality of life [28].

Before each stage of the child's development of ASD, parents face a series of unknown problems that make it difficult to adapt to school: how the child is affected by changing environment, colleagues and teachers, how they perceive atypical behavior, and how it is affected the school year. The major changes in the life of a student with ASD are vulnerable moments. The fact that academic performance, no matter how high, does not ensure a degree of independence for the child can chronicle the stress of his family and endanger the child's school inclusion [29,30]. Other challenges that children with ASD face are social isolation, rejection, lack of social and psychological support [31]. Cognitively, the child diagnosed with autism spectrum disorders is at the level of his age and sometimes even higher. However, emotionally, the children have difficulty relating to those around them, or managing emotions, aspects that make them vulnerable in front of colleagues [32,33]. Therefore, the school inclusion of students with ASD must focus on overcoming these problems. It was found that research conducted after 2012 in the field of education focused on diagnosing children with ASD and their school inclusion. Prior to this period, the concerns of specialists were focused on their families, on the mothers of children with autism [34].

School inclusion programs focus on the inclusion of students with special educational needs (SEN) in regular schools, alongside typical students for academic development and for acquiring social and emotional skills [35].

Carrying out educational projects in which contact-based education (CBE) sessions are carried out increases the level of knowledge and awareness of the difficulties of adapting to the school environment and of community acceptance of students with ASD. Education through contact is an effective way to reduce stigma and discrimination against SEN students and their school inclusion [36–38]. Experts in the field need to be increasingly aware of the need for strategies relevant to the inclusive education of children with disabilities, empirically supported by evidence-based practices [39,40]. For example, a certain choice of classroom management strategies (CMS) by teachers in the teaching program may reduce the impact of mental disabilities, such as ADHD, on school performance [41]. The use of information and communication technology (ICT) in learning processes by teachers can be an effective method of educational inclusion of students with disabilities. The use of ICT is not only a method imposed by the circumstances of the COVID-19 pandemic but can be a key factor in sustainable educational programs for students with specific needs [42,43].

*1.3. The Situation of Children with ASD in Romania*

In Romania, there is no clear evidence of the number of children diagnosed with ASD. The Help Autism Foundation (NGO) estimates for 2019 a number of over 30,000 children with ASD, while over 1000 children are diagnosed annually (according to data provided by the Ministry of Health, National Institute of Mental Health). Of these, more than 10,500 children were of school age: 45% went to boarding schools, 30% to special schools, and 25% were not included in any form of education [44]. At the moment, in Romania, a series of measures have been implemented through national policies for equal opportunities, prevention and treatment of people with disabilities. School inclusion measures have been proposed to target access to education for all children and children with SEN, including

children with ASD. In line with the observance and encouragement of the principles and practices recommended by the Salamanca Declaration (1994), the implementation of educational programs by different educational systems must take into account the diversity of characteristics and learning needs of each child [45].

In Romania, the implementation of policies for the inclusion of students with SEN was carried out in several stages: reorganization and functioning of the special education system and promotion of measures for the integration of students with SEN. In the medium and long term, there is no national strategy for these students. There is also, in addition to the chronic lack of specialists and support teachers, the lack of teaching materials, the inadequate curricula or one that is not alighted with what is required for children with special needs, with an emphasis on the amount of information accumulated rather than on the development of skills, focused on the pace of each child. A real problem for students with SEN is the transition from one schooling cycle to another (especially in high school), where the efforts made by teachers in the previous cycle are not continued, counseling activities are more frequent and not therapeutic activities, and more so those in urban schools. A real need for the inclusion of students with SEN is their involvement in extracurricular activities, activities that do not take place at all (even before the COVID-19 pandemic) [46]. The reasons why teachers do not accept students with SEN in the classroom range from the unpredictability of children and the academic level considered low to the lack of information, specialized support or motivation for teachers and their extra effort. A study conducted in 2015 on a sample of 129 parents who have children with ASD showed that more than half of them were discriminated against and stigmatized in public spaces by strangers or even acquaintances (relatives, friends or neighbors); over two thirds of the interviewed parents were victims of ill treatment in the community at least once in their lives, and their children were victims of aggression. In the school space, teachers ignored or isolated students with ASD (in the study—18%), and their classmates verbally or physically assaulted them (24%). Discriminatory attitudes and behaviors were directed not only at children with ASD but also at parents, a situation that can lead to family isolation, with low chances of integration into the social and school community [47,48].

The school inclusion of children with ASD (SEN) involves the implementation of educational programs focused on those special requirements, on the elimination of discriminatory attitudes and the creation of supportive school communities. The participation of parents, the community, organizations for people with disabilities in the planning and decision-making process on resources for children with SEN is encouraged [49]. For a student with ASD, with relationship and communication deficiencies, the hostile atmosphere in the school space can have devastating effects. It can accentuate self-isolation or even regression, being antagonistic to the school approach of personal, professional, economic and social development of the individual through education, in order to fully realize the child's potential and to participate to a greater extent in society [28].

### 1.4. The Present Study

Education and raising public awareness of the needs of people with psychosocial disabilities reduces social stigma and discrimination, strengthens cohesion and creates an environment conducive to personal development and social inclusion. At the same time, inclusion reduces social costs in the medium and long term [26,28]. Students with ASD included in public education benefit from the advantages of development in the three areas with deficit: social, behavioral and cognitive skills [30,33].

This study presents a school inclusion project for children with ASD, the SucCESS Project—Support for Children with Special Educational Needs and their families (in Romanian—Suport pentru Copiii cu Cerințe Educaționale Speciale, SucCES) where training sessions on CBE were held in order to reduce discrimination and stigmatization, raise awareness and increase the school inclusion of children with ASD or SEN.

The aim of the research was to obtain scientific evidence based on statistical results on the school inclusion of children with ASD in public education, reducing stigma and dis-

crimination through CBE, in order to create supportive communities in the pre-university school environment. The creation of supportive communities is necessary both for the recovery and training of children with ASD as active members of the community and for optimizing the social behavior of neurotypical children. The study started from the following research questions:

1.  What is the level of discrimination and stigmatization of people diagnosed with ASD among typical high school students?
2.  To what extent does carrying out CBE activities reduce discrimination and help the sustainable school inclusion of students with ASD in the public education system?

## 2. Materials and Methods

### 2.1. Participants

The researches carried out within the "SucCESS Educational Project!" took place between October 2018 and April 2019 on a number of 141 participants (total group—TG), 5 classes of high school students (14–18 years) and teachers, from a city in the southern region of Romania, Calarasi, a city located 150 km from Bucharest (with a population of 65,181 inhabitants) [50].

To carry out the research, 2 groups were created: the control group (CG) consisting of 103 subjects (97 students and 6 teachers) and the active group (AG) consisting of 38 subjects (35 students and 3 teachers, which included a schoolchildren diagnosed with ASD). Although the project addressed a specific age segment, high school students also participated in the research and teachers who teach those classes of students. The average age was M = 17.02 years; SD = 4.6, where 93.6% of participants were under 18 years of age. The gender division maintained relatively equal proportions both in general and in the two groups: AG and CG. The ratio between students (under the age of 18) and adult participants (teachers) remained the same in each group: 91.4% in the AG and 93.8% in the CG (Table 1).

**Table 1.** Characteristics of population.

| Characteristics | | Total Group | | Control Group | | Active Group | |
|---|---|---|---|---|---|---|---|
| | | N | % | N | % | N | % |
| No Subjects | | 141 | 100 | 103 | 60.00 | 38 | 40.00 |
| Gender | Male | 58 | 41.1 | 43 | 41.70 | 15 | 39.50 |
| | Female | 83 | 58.9 | 60 | 58.30 | 23 | 60.50 |
| Residence Provenance | Urban | 83 | 58.9 | 67 | 65.00 | 16 | 42.10 |
| | Rural | 58 | 41.10 | 36 | 35.00 | 22 | 57.90 |
| Age | M SD | 17.20 | 4.60 | 16.86 | 3.87 | 17.45 | 7.05 |

In high schools where the research was conducted, the agreement of the directors, teachers and students was requested and obtained. For the AG, the written agreement of the school management was requested and obtained, and the parents (or legal representatives) signed an informed agreement form.

The reason for choosing the city of Calarasi: it is one of the small towns in Romania, and no information and awareness campaigns have been carried out here on people diagnosed with ASD, neither in the community nor at the level of the school population. Families with affected children have to go to a specialist at about 150 km. distance, in Bucharest, because there is a crisis of doctors specializing in child neuropsychiatry [51]. The lack of institutions (public or private) to provide specific therapeutic services and interventions for children with ASD should also be mentioned. At the city level, there is only one public institution, from the social protection system of the child, which offers this type of services only for the children from the social protection system.

### 2.2. Research Procedure and Strategy

The CG consisted of students from 4 classes from different high schools in the city of Calarasi (Barbu Stirbei National College, Mihai Eminescu Theoretical High School, Sandu Aldea Technological High School and Economic College).

The AG consisted of students from a class from another high school (Danubius High School in Calarasi) as well as teachers who took part, together with students, in the project activities. In order to carry out CBE activities, we needed a class of students in which a student with ASD was included. Thus, the students in that class were selected for the AG. Taking into account the age of the students in the AG, we selected four other classes of students for CG, of the same age, from four different high schools in the same locality, to compare the results. The most important high schools in the city were selected. Being a small town, we chose representative high schools. The application of the tools and the development of the project activities were done in the classes for which the directors gave their consent. The teachers who participated in the research were few, as they were those who had the role of supervisors. They were assessed with the same tools and participated in the same activities as the students. Their proportion was small and preserved in both groups (CG and AG).

Carrying out the research in several high schools ensured the objectivity from the perspective of the diversity of the respondents (age, sex, environment of residence and educational level). The two groups (AG and CG) received the Haghighat's standardized stigmatization questionnaire (SSQ1) [25], adapted to the two moments: the initial questionnaire (IQ was SSQ1) and the final questionnaire (FQ, was the modified SSQ1) (Appendix A) which was applied at equal intervals. The two groups (active and control) were created to comparatively evaluate the results obtained after carrying out the CBE activities from the SucCES Project in the classroom in which there was also a student with ASD. Before completing the IQ, participants received a "Social Interaction Guide" that provided information about ASD, because most students lacked the experience of interacting with such a person, taking care that the information did not influence the participants' responses. The guide presented recommendations on how to interact and communicate with a person with ASD, the perspective of people with ASD on social reality, their interaction with other people or how to perceive a task to be performed. Examples of situations presented to students: awareness of the difference in the use of verbal expressions to the detriment of others: "I don't want to"—I choose not to . . . ; "I can not"—I cannot make it (to do things like you) . . .

Below is an example used in the Guide: "The perspective of a person with autism in a crowded place: My hearing may be very sharp. Dozens of people speak fast at the same time. The speakers loudly announce the offer of the day. The music sounds loud. Cash registers beep and squeak, a roaring coffee grinder. The meat slicer creaks, the little ones cry, the strollers squeak, the fluorescent lights buzz. My brain can't filter all this information and I'm overwhelmed! I could have an extremely sensitive sense of smell. The fish on the counter is not fresh, the man next to us did not take a shower today, the delicacy district offers sausage samples, the baby in front of us has a dirty diaper, in turn 3 employees wash the pickles spread on the floor. I feel like throwing up. And so many things disturb my eyes! Fluorescent light is too strong and flickering. Space seems to be moving. The flashing light makes everything around jump and distorts what I see. For me, there are too many objects to focus on (my brain could compensate with "tunnel vision"), the fans spin on the ceiling, and there are so many silhouettes in constant motion. All of this affects the way I feel just sitting there and I no longer have a sense of my own being in space. There is a difference between "I don't want" (I choose not to . . . ) and "I can't" (I cannot manage/I am not able)" [52].

In the demographic data collection vignette, the section was introduced about possible interaction with people with physical and mental disabilities and, specifically, with people diagnosed with ASD. In this vignette, information such as age, sex, place of residence, level of education and occupation were requested. In the two questionnaires (IQ and

FQ), the data in the vignette were modified: students related differently to people with ASD. In IQ, the strategy of assessing the behavior of other people and not the students' was adopted: "What do you think most people (not you, but others) would do if they found out about the diagnosis of ASD?" In FQ, the vignette presented information about the student diagnosed with ASD, and the evaluation focused on the student's behavior: "What do you think you would do if you found out about the diagnosis of ASD?" Both vignettes provided information on how to complete the questionnaire. Students in the AG participated for three days in CBE sessions, through which they were provided with knowledge and different work tools. The group of students also included a student with ASD, who participated with colleagues in all activities. The questionnaires (IQ and FQ) had a number of 13 questions with four identical answers each, the Likert model: "Yes, absolutely"; "Probably yes"; Probably not"; and "Not at all". When calculating the score obtained, the least favorable answers received 1 point, and the most favorable, 4 points. The grouping of the answers into stigmatizing (those with 1 and 2 points) and nonstigmatizing (those with 3 and 4 points) was also used, thus making the interpretation easier. To avoid ticking the answers without reading the question, the answers were arranged in such a way as to alternate, so as not to place all the answers with the same score on the same side of the questionnaire. For the translation into Romanian of the questionnaires, both the English version and the Italian version were used [25,26], and specialists were consulted for both variants. When translating the final questionnaire, the negative form was preferred to some questions ("you would dislike . . . ") (Appendix B, the Romanian version of the questionnaire). We modified the questions, because it was considered that their affirmative form ("would you like . . . ") presupposes a voluntary approach on the part of the respondent and thus conditions the answer (Figure 1).

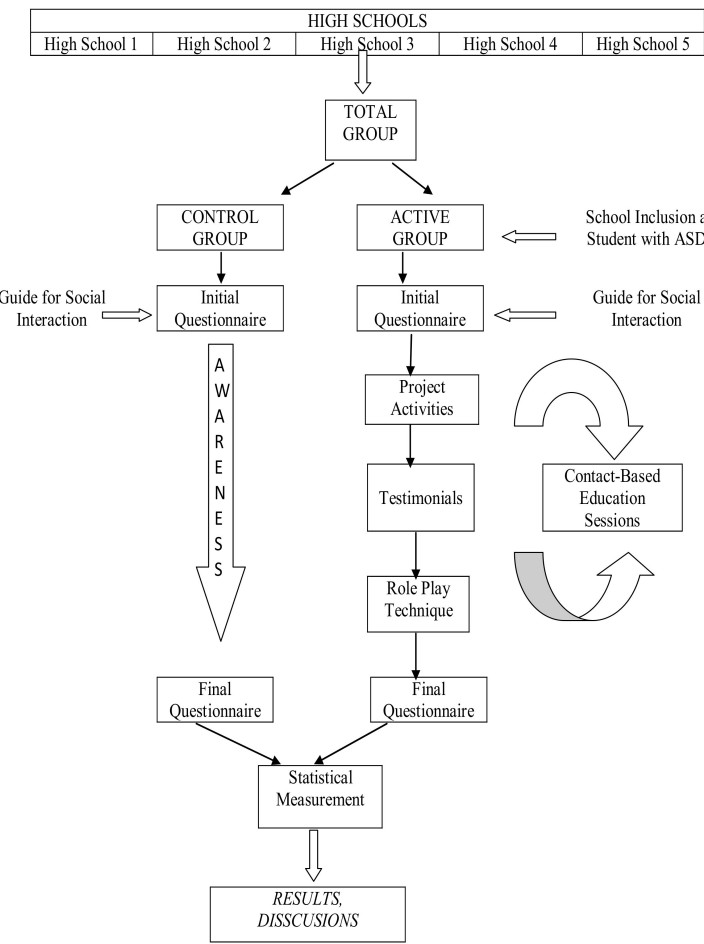

**Figure 1.** Strategy of the research.

*2.3. Measures*

The data collected in the research were processed using the statistical package for the social sciences (SPSS) software, version 20. The results thus obtained were compared between the two groups but also within each group, using different correlations. In the AG, the data resulting from the application of the Haghighat's SSQ1 pre-activity and post-activity CBE questionnaire from the SucCESS Project were compared. All the questionnaires completed by the students and teachers can be found at the headquarters of the "SucCESS Association! Support for children with special educational needs and their families", Calarasi. The project included CBE sessions: watching video/testimonial materials, games, workshops, role play technique, free discussions and other group activities, which followed the general purpose of the project. Any learning process involves the sharing of experiences in the field studied, necessary to fix the information presented. For this reason, in addition to exposing the topic in question, games and flyers, three representative videos/testimonials for the purpose of the project were also presented to the students (Appendix C). The selection of the three videos focused on the informative value and the emotional side, raising the audience's awareness of the problems of people with ASD.

## 3. Results

The application of the questionnaires led us in the initial stage, to identify discriminatory and stigmatizing attitudes in high school students and the ability to interact with a person diagnosed with ASD. Internal consistency and fidelity were verified by calculating the Cronbach alpha coefficient for the two questionnaire variants (IQ and FQ) applied to the two groups. Therefore, for the AG: IQ-Cronbach's $\alpha$ = 0.813 (M = 2.81); FQ-Cronbach's $\alpha$: 0.735 (M = 3.55); for the CG: IQ-Cronbach's $\alpha$ = 0.755 (M = 2.72); FQ-Cronbach's $\alpha$ = 0.768 (M = 3.26) were the significant values that supported the approach.

The demographic vignette provided us with information that was correlated with the intention to interact with people with physical, mental and ASD disabilities. The information was relevant because the interaction with a person with a disability requires a certain degree of knowledge of the problems they face and, implicitly, a different attitude from those who do not interact with people with disabilities. The analysis of the answers shows that, in general, students did not interact to a large extent with people with physical or mental disabilities (Table 2.). Compared between the two types of disabilities, students interacted more with people with mental disabilities (36%). The exception is the AG, which includes a student with ASD.

**Table 2.** Students' interaction with people with different types of disabilities.

| | The Interaction with People with | | | | | | | | |
|---|---|---|---|---|---|---|---|---|---|
| | **Physical Disabilities (%)** | | | **Mental Disabilities (%)** | | | **Diagnosed with ASD (%)** | | |
| Groups | TG | CG | AG | TG | CG | AG | TG | CG | AG |
| daily | 17 | 13.6 | 47.4 | 7.8 | 4.9 | 15.8 | 17 | 5.9 | 47.4 |
| occasional | 24.1 | 53.4 | 10.5 | 36.2 | 33 | 44.7 | 24.1 | 29.1 | 10.5 |
| never | 58.9 | 33 | 42.1 | 56 | 62.1 | 39.5 | 58.9 | 65 | 42.1 |

The comparison of the results obtained by both groups by applying IQ and FQ shows a significant change in discriminatory attitudes toward people with disabilities in both groups but especially in the AG. These changes aimed at reducing the degree of stigma and discrimination that came from awareness of the problems faced by people with disabilities and from the way questions are asked. The way questions are asked can influence the answer in the sense that when people are asked directly about their opinion on a problem, they automatically go on the defensive and seek to give an answer that benefits them [23]. Thus, in IQ, I asked the opinion about the attitude of other people regarding the analyzed problem: "Do most people think that this person missed out on life?".

In FQ, the questions are addressed directly to the student: "Do you think this person missed out on life?", and we considered that it already has the exercise of completing the

questionnaire. There is a decrease in the least favorable answers −1 point: from 10.45% (IQ) to 3.95% (FQ) and a doubling of the most favorable answers −4 points: from 24.26% (IQ) to 49.96% (FQ) (Figure 2).

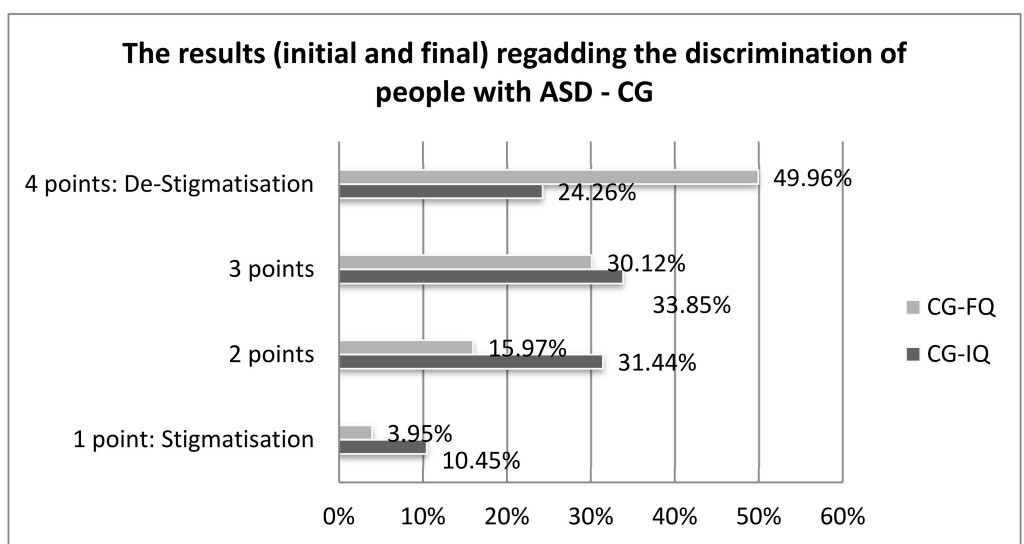

**Figure 2.** The results (initial and final) regarding the discrimination of people with ASD—control group (CG).

The CBE activities supported by the AG changed the discriminatory attitudes toward their decrease by significant percentages: after the application of FQ there were no more 1-point answers, the most stigmatizing (0%), compared to the initial assessment stage (IQ application) where the answers were 8.5% (Figure 3). Moreover, the number of 4-point answers, the most favorable increased greatly: from 12.75% in IQ to 62.54% in FQ.

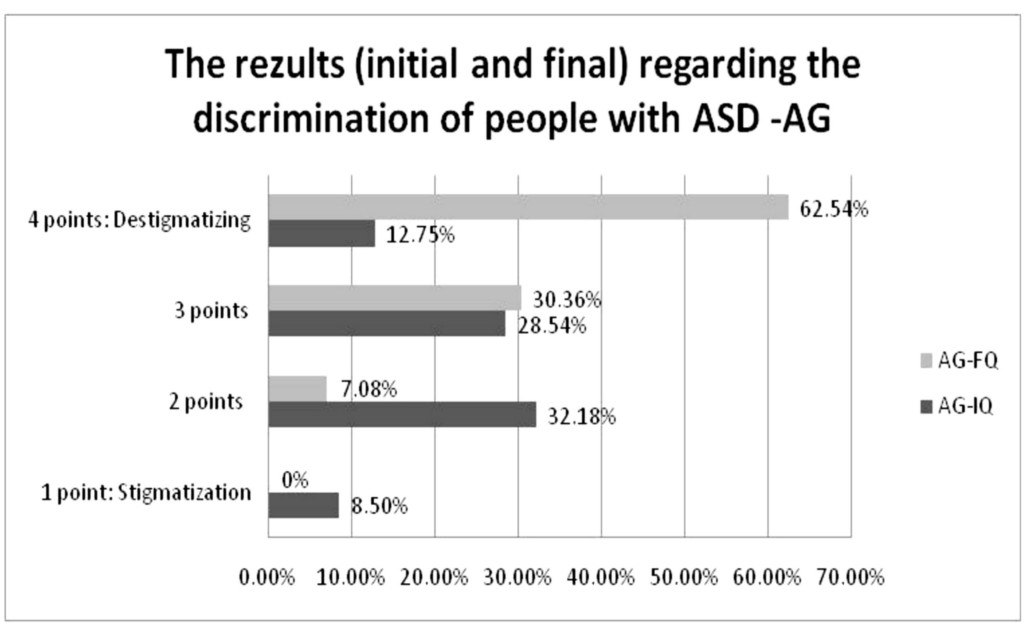

**Figure 3.** The results (initial and final) regarding the discrimination of people with ASD—active group (AG).

There are differences between the two groups right from the first stage (IQ application) comparing the data between them:

a.  the most stigmatizing answers, of 1 point, were approximately equal: in CG-10.45% and in AG-8.50%;
b.  the most favorable answers, of 4 points, were different, higher for CG: of 24.26% compared to 12.74% for AG.

After the application of FQ, greater changes were observed between the attitudes of students who participated in the activities of the SucCES project and those who did not have direct contact with students with ASD:

a.  the most stigmatizing answers, of 1 point, disappeared completely in AG, 0%, compared to CG-3.95%;
b.  favorable answers increased more in the AG than in the CG: 62.54% compared to 49.96% (Figures 2 and 3).

## 4. Discussion

Analyzing the answers from the perspective of the students' residence environment, applying IQ and FQ, we observe the decrease in the number of stigmatizing and discriminatory answers of 1 and 2 points (Table 3). In comparison, it is observed that respondents who have permanent residence in urban areas had less discriminatory responses than students in rural areas. If we group the answers in two categories, stigmatizing-the answers of 1 and 2 points and nonstigmatizing, those of 3 and 4 points, we find that the students from the urban environment had higher percentages in 6 of the 8 groups of answers of 3 and 4 points-nonstigmatizing.

**Table 3.** Relationship between discriminatory attitudes and environment of residence.

| Score Obtained | Control Group (CG) (%) | | | | Active Group (AG) (%) | | | |
|---|---|---|---|---|---|---|---|---|
| | IQ | | FQ | | IQ | | FQ | |
| | Urban | Rural | Urban | Rural | Urban | Rural | Urban | Rural |
| 1p | 9 | 14 | 3 | 6 | 11 | 9 | 0 | 0 |
| 2p | 33 | 29 | 13 | 22 | 29 | 34 | 7 | 7 |
| 3p | 34 | 33 | 30 | 29 | 29 | 28 | 34 | 28 |
| 4p | 24 | 24 | 54 | 43 | 31 | 29 | 59 | 65 |

After the application of IQ, there is only one higher score in the CG for rural students, for the answers of 1 point-14%, compared to 9% for urban students, and for the values of 4 points (the least discriminating), no differences were registered, and the answers were identical-24%. After the application of FQ to AG, the percentages of stigmatizing responses (1 and 2 points) changed significantly, decreased, and showed no differences between rural and urban (0 and 7%). Large differences are observed in rural students in AG-65%, compared to those in CG, 43%, which shows that the CBE sessions were efficient.

When distributing the answers according to the students' gender, the answers were relatively homogeneous, without significant differences: some of the pairs of answers, both discriminatory and least discriminatory, recorded equal levels or differences of only one percent. Male respondents recorded higher percentages in 5 of the 8 situations, compared to only 3 of female respondents; however, there are very small differences, which shows a high homogeneity of responses (Table 4).

The obtained results show that the stigma of students with SEN/ASD can be reduced by CBE activities that help to create supportive communities in the school space. The results obtained in the two groups confirmed Haghigat's unitary theory of stigmatization [25] which presents the social, psychological and evolutionary foundations of stigmatization. Del Cassale [26] used Haghighat's scale (SSQ) to assess the stigma of people with severe mental illness among a school population (16–18 years old). The students followed an educational program to improve the stigmatizing perception, and the effectiveness of awareness activities and reduction of stigma were observed statistically. The results obtained by Del Cassale showed that the lack of information about the problems

and suffering of people with mental disorders, causes social stigma and discrimination, and through educational activities can positively change attitudes toward this social category. Other authors [36,37] have used strategies to reduce stigma against mental illness in health students by including contact education sessions in their educational programs. The research findings showed that the interaction of students (trainees and future health professionals) with people diagnosed with mental illness in these education sessions is an effective method of reducing stigma. Morgieve M. et al. [38] in a research (The CrazyApp survey) on the study of representations and stigmatizing attitudes toward people with mental illness conducted in France, used technology (smartphone or computer) to perform certain contact-based interventions. Videos were presented, short testimonials of people with certain mental problems (<2 min), followed by the application of questionnaires that evaluated the participants' answers. It has been observed that such interventions improve classical strategies to reduce stigma. In our study, we combined these methodologies and research tools that were identified in other studies; we applied the SSQ1 questionnaire, used videos, testimonials of some people, and included a student diagnosed with ASD, who participated with colleagues in project activities. We believe that the use of a mix of methods in CBE activities can contribute substantially to the school inclusion of students with SEN, a methodology that we have not found in other previous studies.

**Table 4.** The relationship between discriminatory attitudes and students' gender.

| Score Obtained | Control Group % | | | | Active Group % | | | |
| --- | --- | --- | --- | --- | --- | --- | --- | --- |
| | IQ | | FQ | | IQ | | FQ | |
| | M | F | M | F | M | F | M | F |
| 1p | 11 | 10 | 5 | 3 | 10 | 8 | 0 | 0 |
| 2p | 31 | 32 | 20 | 12 | 26 | 35 | 7 | 7 |
| 3p | 34 | 33 | 32 | 29 | 33 | 27 | 32 | 29 |
| 4p | 24 | 25 | 43 | 56 | 31 | 30 | 61 | 64 |

## 5. Conclusions

Reducing stigma and discrimination can also be achieved in the school-age population, arguments that support the initiation of educational programs for students through which perceptions about people with ASD can be changed. By changing the attitudes of the school community, we produce changes, and at the level of the attitudes of the community members, we have a higher degree of acceptance, with advantages on both sides [53]. The SucCESS Educational Project has led to awareness among high school students of the problems faced by people with ASD, with SEN in general, and has been statistically demonstrated to significantly reduce discriminatory and stigmatizing attitudes after CBE. The use of the SSQ1 questionnaire [25] before and after CBE activities surprised the change in discriminatory attitudes toward people with mental disabilities in high school students. The acquisition of new information about ASD and new skills to communicate and interact with students with ASD/SEN confirmed that they can take place information and awareness campaigns on their suffering and school exclusion. Lack of knowledge before the CBE shows the direct relationship between discrimination, stigma and the acquisition of new knowledge.

The realization of the SucCESS Educational Project through CBE sessions led to the development of a supportive community in the AG regarding the students with SEN in the school space. Creating supportive communities first in school and then in society as students grow and become adults reduces discrimination and stigmatization of people with ASD or other disabilities at the public level. Communities do not in within themselves change their discriminatory attitude toward people with disabilities, even if there are laws that create the theoretical framework for limiting differences and giving equal opportunities socially and educationally. That is why there is a need, at school level, at all levels, for educational programs aimed at increasing the degree of school and community inclusion

for people with special needs, objective pursued in the SucCESS Educational Project! (Support for Children with Special Educational Needs).

## 6. Limitations and Future Lines of Research

A continuation of the present research can go in the direction of monitoring the effects created by the activities carried out with the AG to determine if the SucCESS Project is an effective tool that can be replicated at other educational levels, for example gimnasial school, with a population aged 10–15 years. Although the project conveys concepts that sometimes go beyond the possibilities of understanding and assimilation of children in gimnasial school, some of the activities can be adapted for this age group. As the CBE unfolded, the students became more interested in the project activities, asking questions about the analyzed subject, an interest that was not displayed in the beginning. After the videos/testimonials, they became more curious and attentive to the activities carried out in class together with the colleague diagnosed with ASD. However, these relationships will be analyzed in future research, and we do not just accept simple assumptions.

The SucCESS Project also involved carrying out extracurricular activities, socializing and participating in artistic activities, but due to the COVID-19 pandemic, school activities were suspended and moved to the online, and we will apply them in future stages.

The COVID-19 pandemic also affected the education system in Romania; the courses were conducted online or through television (for example, Tele-school). Children with SEN were negatively affected by the suspension of courses, and those with ASD no longer had access to specialized therapies, which led to the stagnation of development and even the loss of acquisitions accumulated by children in the period before the pandemic. The parents had to access the therapy services by phone and, despite the lack of specialized training, to supplement the absence from therapy with specific activities, for which they relied, more than ever, on the guidance given by telephone by therapists. Lack of interaction with people outside the nuclear family circle can increase the child's level of isolation, as well as the sudden and massive increase in the level of stress experienced by parents due to social isolation, which are aspects that make them vulnerable [22,23]. Social isolation has induced behavioral changes difficult to manage by parents and children without neurobehavioral disorders, and it is obvious that in the case of children with neuro-mental disabilities can occur complex pathologies, difficult to recover and correct [54]. Returning to into classes with the physical presence of students and teachers in the classroom will be a challenge for all actors involved, to recreate the necessary infrastructure to resume school activities in good condition. Behavioral changes should not be understood as possible and rare but should be considered from the beginning and for which it is necessary to act prophylactically and not after the negative effects have taken shape and are visible. Educational programs such as SucCESS Project could have the ability to prevent some of the negative effects of lack of interaction in the school environment during the pandemic, with benefits for both students with SEN and other students, teachers and the entire school community.

**Author Contributions:** Conceptualization, G.N. and C.-L.D.; methodology, G.N. and C.-L.D.; investigation, C.-L.D.; formal analysis, G.N. and C.-L.D.; data curation, G.N.; writing—original draft preparation, C.-L.D.; writing—review and editing, G.N.; visualization, G.N. and C.-L.D.; supervision, G.N. All authors have read and agreed to the published version of the manuscript.

**Funding:** This research received no external funding.

**Institutional Review Board Statement:** Not applicable.

**Informed Consent Statement:** Informed consent was obtained from all subjects involved in the study.

**Data Availability Statement:** Data are contained within the article.

**Conflicts of Interest:** The authors declare no conflict of interest.

## Appendix A

*Haghighat's Standardized Stigmatization Questionnaire (SSQ1)-Final Questionaire (FQ)*
Age: | _ | _ | Sex: | _ | Occupation: _______________ Residence: ◯ urban ◯ rural
I interact with people diagnosed with ASD      ◯ daily ◯ several times a year ◯ never
I interact with people with disabilities:physical ◯ daily ◯ several times a year ◯ never
I interact with people with disabilities:(other than ASD)-psychic ◯ daily ◯ several times a year ◯ never
Education level: ◯ Gymnasium ◯ High school ◯ University ◯ Postgraduate studies

*Thank you for choosing to answer a few questions about a person who has a medical diagnosis. Your name is not required for this search. Please read carefully and then answer the questions about this person. There are many opinions on this subject. There are no right or wrong answers: we are interested in your opinion. Please answer each question. Your name is not required.*

*The person in question is 15 years old, is a schoolchildren and lives with his parents. Following a medical investigation, he was diagnosed with Autism Spectrum Disorder (ASD). What do you think you would do if you found out about this diagnosis? Please tick your answer to each of the 13 questions.*

| 1. You would dislike to sit next to this person on a bus? | | | |
|---|---|---|---|
| a. Yes, absolutely | b. Probably yes | c. Probably not | d. No, not at all |

| 2. You would dislike to eat food cooked by this person? | | | |
|---|---|---|---|
| a. Yes, absolutely | b. Probably yes | c. Probably not | d. No, not at all |

| 3. You would avoid talking with this person if possible? | | | |
|---|---|---|---|
| a. Yes, absolutely | b. Probably yes | c. Probably not | d. No, not at all |

| 4. Do you think that this person should spend his whole life in a hospital or other care institution? | | | |
|---|---|---|---|
| a. Yes, absolutely | b. Probably yes | c. Probably not | d. No, not at all |

| 5. You would dislike that this person to become the teacher of their children? | | | |
|---|---|---|---|
| a. Yes, absolutely | b. Probably yes | c. Probably not | d. No, not at all |

| 6. Would you dislike this person becoming a relative to you through an alliance (marriage to a member of your family)? | | | |
|---|---|---|---|
| a. Yes, absolutely | b. Probably yes | c. Probably not | d. No, not at all |

| 7. would you dislike to work / learn with this person? | | | |
|---|---|---|---|
| a. Yes, absolutely | b. Probably yes | c. Probably not | d. No, not at all |

| 8. Would you be scared if this person moved / lived in your neighborhood? | | | |
|---|---|---|---|
| a. Yes, absolutely | b. Probably yes | c. Probably not | d. No, not at all |

| 9. Do you think that the main causes of this person's illness are lack of moral strength or will power? | | | |
|---|---|---|---|
| a. Yes, absolutely | b. Probably yes | c. Probably not | d. No, not at all |

| 10. Do you think that this person's affection is a punishment for the bad deeds he has committed? | | | |
|---|---|---|---|
| a. Yes, absolutely | b. Probably yes | c. Probably not | d. No, not at all |

| 11. Do you think that this person is hiding behind the condition to avoid the difficulties that everyday life entails? | | | |
|---|---|---|---|
| a. Yes, absolutely | b. Probably yes | c. Probably not | d. No, not at all |

| 12. Do you think this person missed their life? | | | |
|---|---|---|---|
| a. Yes, absolutely | b. Probably yes | c. Probably not | d. No, not at all |

| 13. Could you think this person is a bad man? | | | |
|---|---|---|---|
| a. Yes, absolutely | b. Probably yes | c. Probably not | d. No, not at all |

# Appendix B

Haghighat's SSQ1 – Final Questinaire, Romanian Version/ *Chestionarul standardizat de stigmatizare (Haghighat R, 2005)- Chestionarul final  (C.F.)*

Vârstă: |_|_| Sex: |_| Ocupaţie: _________________________  Mediul de reşedinţă:  ○ urban   ○ rural
Interacţionez cu  persoane diagnosticate cu TSA:  ○ zilnic   ○ de câteva ori pe an   ○ niciodată
Interacţionez cu persoane cu dizabilităţi:  fizice   ○ zilnic   ○ de câteva ori pe an   ○ niciodată
 ................cu dizabiităţi psihice, altele decât TSA: ○ zilnic   ○ de câteva ori pe an   ○ niciodată
Nivelul de educaţie absolvit: ○ Gimnaziu   ○ Liceu   ○ Universitare   ○ Studii postuniversitare

　　　　Îţi mulţumesc că ai ales să răspunzi câtorva întrebări despre o persoană care are un diagnostic medical. Numele tău nu este necesar pentru această cercetare. Te rog să citeşti cu atenţie şi apoi să răspunzi la întrebările despre această persoană. Există multe opinii pe marginea acestui subiect. Nu există răspunsuri corecte sau greşite: pur şi simplu mă interesează opinia ta. Te rog să răspunzi la fiecare întrebare. Numele tău nu este necesar.*Persoana în cauză are 15 ani, este  elev şi locuieşte cu părinţii săi. În urma unei investigaţii medicale, a primit  diagnosticul de **tulburări din spectrul autist (TSA)**.Ce crezi că ai face tu dacă ai afla despre acest diagnostic?* Te rog să bifezi răspunsul tău la fiecare din cele 13 întrebări.

| 1. Ţi-ar displăcea să stai lângă această persoană într-un autobuz? | | | |
|---|---|---|---|
| a. Da, absolut | b. Probabil că da | c. Probabil că nu | d. Nu, deloc |

| 2. Ţi-ar displăcea să mănânci mâncare gătită de această persoană? | | | |
|---|---|---|---|
| a. Da, absolut | b. Probabil că da | c. Probabil că nu | d. Nu, deloc |

| 3.  Ai evita să vorbeşti cu această persoană dacă ar fi posibil? | | | |
|---|---|---|---|
| a. Da, absolut | b. Probabil că da | c. Probabil că nu | d. Nu, deloc |

| 4. Consideri că această persoană ar trebui să-şi petreacă toată viaţa într-un spital sau o altă instituţie de îngrijire? | | | |
|---|---|---|---|
| a. Da, absolut | b. Probabil că da | c. Probabil că nu | d. Nu, deloc |

| 5.  Ţi-ar displăcea ca această persoană să devină profesorul copiilor lor? | | | |
|---|---|---|---|
| a. Da, absolut | b. Probabil că da | c. Probabil că nu | d. Nu, deloc |

| 6.  Ţi-ar displăcea ca această persoană să devină rudă cu tine prin alianţă (căsătorie cu un membru al familiei tale)? | | | |
|---|---|---|---|
| a. Da, absolut | b. Probabil că da | c. Probabil că nu | d. Nu, deloc |

| 7. Ţi-ar displăcea să lucrezi/înveţi împreună cu această persoană? | | | |
|---|---|---|---|
| a. Da, absolut | b. Probabil că da | c. Probabil că nu | d. Nu, deloc |

| 8. Ai fi speriat dacă această persoană s-ar muta/ar locui în vecinătatea ta? | | | |
|---|---|---|---|
| a. Da, absolut | b. Probabil că da | c. Probabil că nu | d. Nu, deloc |

| 9. Crezi că principalele cauze ale afecţiunii acestei persoane sunt lipsa de voinţă sau de moralitate? | | | |
|---|---|---|---|
| a. Da, absolut | b. Probabil că da | c. Probabil că nu | d. Nu, deloc |

| 10.  Crezi că afecţiunea acestei persoane este o pedeapsă pentru faptele rele pe care le-a săvârşit? | | | |
|---|---|---|---|
| a. Da, absolut | b. Probabil că da | c. Probabil că nu | d. Nu, deloc |

| 11 Crezi că această persoană se ascunde în spatele afecţiunii pentru a evita dificultăţile pe care le presupune viaţa de zi cu zi? | | | |
|---|---|---|---|
| a. Da, absolut | b. Probabil că da | c. Probabil că nu | d. Nu, deloc |

| 12 Crezi că această persoană şi-a ratat viaţa? | | | |
|---|---|---|---|
| a. Da, absolut | b. Probabil că da | c. Probabil că nu | d. Nu, deloc |

| 13. Ai putea crede că această persoană este un om rău? | | | |
|---|---|---|---|
| a. Da, absolut | b. Probabil că da | c. Probabil că nu | d. Nu, deloc |

# Appendix C

**Video / testimonial materials used in the SucCES Educational Project**

- Marius Manole şi Andi Vasluianu despre Autism Voice (two Romanian actors about Autism Voice):
  https://www.youtube.com/watch?v=7059At_J8aU

- Autism: a quick trip to my home planet | Monique Botha | TEDxSurreyUniversity:
  https://www.youtube.com/watch?v=NCAErePScO0

- Temple Grandin - The world needs all kinds of minds:
  https://www.ted.com/talks/temple_grandin_the_world_needs_all_kinds_of_minds/up-next; the variant in Romanian: https://www.ted.com/talks/temple_grandin_the_world_needs_all_kinds_of_minds?language=ro#t-502096

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
