# Peer review of "Preventing School Exclusion of Students with Autism Spectrum Disorder (ASD) through Reducing Discrimination: Sustainable Integration through Contact-Based Education Sessions"

_sustainability, doi:10.3390/su13137056_

Round 1
Reviewer 1 Report
vThe object investigated in the article does not agree with the theme of the journal.
The study objective is not well identified in the abstract or in the introduction. I think the authors should specifically introduce it at the beginning of both sections.
The investigated topic is of scientific and social interest.
The article lacks cohesion, especially in the Conceptual Framework section. However, the following sections, in which the study is described and conclusions are drawn, have a correct, well-structured and cohesive design.
The statistical treatment is correct.
The conclusions are well drawn and interesting.
The authors incorporate a section of limitations, which demonstrates their honesty and frankness in the investigation. In my opinion, this section is very convenient and timely.
In general, the article is correct but I consider that the theme is not in line with the research objectives of the journal.
Reviewer 2 Report
An interesting article is presented for the scientific community. A sensitive issue is addressed, social exclusion, in current education. In people with ASD, due to their peculiarities, special attention must be paid to this aspect.
Regarding the elaboration of the manuscript, the following improvement proposals are detailed:
-The title is too long. It is recommended to reduce the number of words.
-The conceptual framework is brief. It is recommended to analyze the following articles:
https://www.mdpi.com/2076-3425/11/1/74
https://www.mdpi.com/2076-3425/10/12/1018
https://www.mdpi.com/2076-3425/10/12/985
-The quality of the figures is very low.
-The authors must check the language as errors have been detected in the writing.
-The results are not discussed with previous studies. Authors should separate the results section from the discussion. Discussion is one of the most important parts of a manuscript. The findings of this study should be discussed with previous research on the state of the art.
-The limitations and future lines should be placed after the conclusions.
Reviewer 3 Report
Preventing school exclusion of students with Autism Spectrum 2 Disorder (ASD) through reducing discrimination. Sustainable 3 integration through contact-based education sessions: SucCES 4 Project - preliminary study is very topical as the phenomenon of discrimination against people with autism occurs more and more frequently despite the implementation of various educational programs.
The text is appropriate for the special Issue area of ​​"Sustainable Education: The Educational Response to Students with Disabilities", however, the research methodology is questionable. It is not known how the selection of the research sample, both active and control (selection of students and schools) was carried out, and with what tools the teachers were assessed. Are they the same as the students (the results of the research covering the surveyed teachers are not described)?
Therefore, the presented research results do not entitle to the presented general conclusions, but only in relation to this group of respondents.
The use of a standardized tool in the research is a great advantage, however, in the section Results and discussion, there is no reference to the results of research by other authors who have already used the standardized tool (perhaps there were only pilot studies conducted earlier?
In their conclusions, the authors indicate that changing attitudes is most effective in childhood and adolescence. However, the understanding of the definition of an attitude itself was not discussed. The best known in education is the one that deals with its three components: cognitive, emotional and behavioral. If so, the correlations between these components should be examined and which of them should be changed or strengthened.
It is also worth describing the preparation of teachers in Romania to work with autistic children, in an integrated form with students without social dysfunctions. Do they have training courses in this area at all? If not, it is worth taking up this issue in further research, because often the reluctance to work with children with various disabilities results from the lack of competence of educators in this area and in preparation for the teaching profession. It is also worth presenting the available research results indicating the reasons for the reluctance of teachers to admit students with ASD to classes with students without such disorders.
Round 2
Reviewer 2 Report
The authors have taken into account the comments offered to improve the manuscript. Therefore, the study has improved its quality with respect to the version initially presented. My decision is to accept this version of the manuscript.